

# Fish Ontology framework for taxonomy-based fish recognition

Najib M. Ali[1], Haris A. Khan[1], Amy Y-Hui Then[1], Chong Ving Ching[1], Manas Gaur[2] and Sarinder Kaur Dhillon[1]

[1] Institute of Biological Sciences, Faculty of Science, University of Malaya, Kuala Lumpur, Malaysia
[2] Wright State University, Kno.e.sis Center, Dayton, OH, United States of America

## ABSTRACT

Life science ontologies play an important role in Semantic Web. Given the diversity in fish species and the associated wealth of information, it is imperative to develop an ontology capable of linking and integrating this information in an automated fashion. As such, we introduce the Fish Ontology (FO), an automated classification architecture of existing fish taxa which provides taxonomic information on unknown fish based on metadata restrictions. It is designed to support knowledge discovery, provide semantic annotation of fish and fisheries resources, data integration, and information retrieval. Automated classification for unknown specimens is a unique feature that currently does not appear to exist in other known ontologies. Examples of automated classification for major groups of fish are demonstrated, showing the inferred information by introducing several restrictions at the species or specimen level. The current version of FO has 1,830 classes, includes widely used fisheries terminology, and models major aspects of fish taxonomy, grouping, and character. With more than 30,000 known fish species globally, the FO will be an indispensable tool for fish scientists and other interested users.

## INTRODUCTION

Increasing amount of data produced on a single species has made it harder for fish researchers to manage and provide fish data in a single database. Moreover, the high demand for metadata for a single species is driving researchers to find a better alternative for the current database structure (discussed further below). The Semantic Web technology, with the capability to give well-defined meaning to information and to enable better cooperation between human and computer, provides a promising platform for biodiversity researchers who are interested to link and share their data in common public repository.

Presently, most of the fish databases are constructed using relational database models, focusing on species related information. Data in these repositories are usually structured based on the researcher's interests and personal needs, which in turn restrict the application of a uniform naming standard. Hence a preferred way to provide species data is in the form of an ontology, a structured vocabulary that describes entities of a domain of interest and their relationships (*Shadbolt, Hall & Berners-Lee, 2006*). A relational database focuses

Corresponding author
Sarinder Kaur Dhillon,
sarinder@um.edu.my

on the data whereas, an ontology provide meaning to the data with the help of metadata. Using metadata, an ontology can be linked and mapped to other related ontologies and this information can be used to automatically infer and recognize the relevant or contextually related result for a given search.

There are several important and popular projects in the fish and fisheries domain developed as conventional back-end databases such as the Catalog of Fishes (*Eschmeyer, Fricke & Van der Laan, 2014*), FishBase (*Froese & Pauly, 2000*), IGFA Fish Database (*International Game Fish Association, 2016*), The New Zealand Freshwater Fish Database (*National Institute of Water and Atmospheric Research, 2016*) , The Fish Database of Taiwan (*Shao, 2001*), Fish Stocking Database (*Great Lakes Fishery Commission, 1997*), FishTraits (*Emmanuel & Angermeier, 2013*), and Fish Barcode of Life (*iBOL Working Group, 2005*). While these databases provide extensive and up to date information on fish, they are not based on ontology and hence do not support Semantic Web deployment unless converted into appropriate formats (*Ankolekar et al., 2007*). Furthermore, most of them are not created based on Semantic Web principles (*Berners-Lee, 2016*) and there is little effort dedicated to create an automated fish species identification using the Semantic Web approach. An ontology can enhance information providing capability for a database by the use of metadata to discover and gather new information from other databases, or by linking them to create a better information network. Thus, the work laid out in this paper is created as an effort to address these problems.

To date, no dedicated ontology with automated classification for fish exists, with the exception of the Network of Fisheries Ontology (NFO) (*Caracciolo et al., 2012*) which focuses on fisheries activity and selected species of commercial interests, and the Marine Top Layer Ontology (MarineTLO) (*Tzitzikas et al., 2016*) which focuses on marine animal. Both of these ontologies are not primarily focused on fish, and they do not possess automated classification capability. Given that the total number of fish species has been estimated at 32,000 to 40,000 globally (*Nelson, 2006*; *Chapman, 2009*; *Eschmeyer et al., 2010*), an automated and comprehensive fish classification platform would be an indispensable tool for fisheries biologists, marine scientists, and even laypeople with interest in fish. Thus, in this paper, we propose a fish-based ontology that is able to automate group classification, and to link terms used by research on the fish domain with related terms from other research domains.

This paper describes the framework of the Fish Ontology (FO) for precise and comprehensive semantic annotation of fish resources (e.g., datasets, documents, and models) where it can be used to fill in the gap of distinct terms which are missing in other ontologies. The FO is an effort to develop and maintain a controlled, structured vocabulary of terms which describe fish anatomy, morphology, ecology and various developmental stages. The FO reuses many terms from other ontologies which are related to and appropriate for the fish and fisheries domain. Additionally some terms such as "Location", "Shape", and "Threat" are implemented to add more description to fish, with the intention to provide more diverse search results.

Originally the FO was developed as a data warehouse for several database formats. It has since evolved to host information on captured and observed fish specimen (e.g., data on

captured samples, captive specimens, or from observational experiments). After undergoing several modifications based on reviews by fish experts, both of these features were merged in the current FO, expanding its functionality to incorporate fish classification and reasoning capability. The FO framework outlined in this paper (current version v1.0.2, Aug 2016) is designed to facilitate integration with related ontologies which is in line with the Big Data Initiative (*IEEE, 2016*) that aims for diverse analytic options.

## METHODS

We used Protégé to create, edit and manage the Fish Ontology and all its terms and relationships (*Musen, 2015*). This open access software contains all the tools needed for this research since it contains sufficient plugins to assist in development and visualization of ontology. Furthermore, Protégé provides several reasoner engines such as Hermit, FaCT++, and Pellet, to provide variation in ontology validation and reasoning (*Tsarkov & Horrocks, 2006*; *Sirin et al., 2007*; *Glimm et al., 2014*). There are also various visualization tools that are provided by Protégé such as OWLVIZ, Ontograf, and VOWL (*Falconer, 2010*; *Horridge, 2010*; *Negru, Lohmann & Haag, 2014*).

The FO is created using Web Ontology Language (OWL) which allows us to query using triple based query languages such as SPARQL (*Prud'hommeaux & Seaborne, 2008*), SPARQL-DL (*Sirin & Parsia, 2007*), and Description Logic (DL) (*Baader et al., 2003*). A triple query (composed of subject–predicate–object) can perform more complex query compared to a relational database query (composed of columns and rows), and is able to retrieve more information due to the Semantic Web architecture which enables them to pull data from URIs or URLs with related metadata (*Alexander, 2013*).

"The Diversity of Fishes: Biology, Evolution, and Ecology" was the main reference used in identifying terms and definitions while devising the FO (*Helfman et al., 2009*). This book is a well-established reference that follows standard fish taxonomy nomenclature proposed by Nelson (*Nelson, 2006*). Most of the class labels, synonyms and definitions in the FO correspond to those in the reference book. Some of the terms for specimen entries are taken from experimental data such as sampling data provided by *Chong, Lee & Lau (2010)*, while others are taken from public online entries such as Wikipedia (*Wikimedia Foundation, 2001*) and DBpedia (*Heath & Bizer, 2011*). We also incorporated classes from other ontologies into our ontology to model the FO classes and enhance its automatic recognition capabilities.

One of the most important aspects in ontology creation is consistency; hence, we sought to follow a standard naming convention while creating the FO. There are no obligatory naming conventions for the creation of OWL classes and properties; however, we decided to use the Camel Case (also known as Camel Back) notation to ensure that the ontology terms and naming are consistent (*Campbell, 2006*; *Horridge et al., 2011*). This naming convention has the advantage of creating more meaningful names by using an expressive sequence of words while respecting the naming constraint (*Horridge et al., 2011*). As such, all of the classes in the FO use the Upper Camel Case notation, while all of its properties use the Lower Camel Case notation. Furthermore some properties are appended with

**Table 1** Statistic of imported or integrated class and properties.

| Ontology or Standard | Number of classes |
| --- | --- |
| Zebrafish Anatomy and Stage Ontology (ZFA, ZFS) | 2 |
| Darwin Core | 2 |
| Vertebrate Taxonomy Ontology (VTO) | 1,345 |
| NCBI organismal classification (NCBITaxon) | 13 |
| Total | 1,362 |

the prefixes of 'has', or 'is', as per the convention recommendation (e.g., "hasBodyPart", "isPartOf"). This naming convention helps clarify the properties to human and to some tools in Protégé (e.g., The "English Prose Tooltip Generator" which uses this naming convention to generate more human readable expressions for class description).

As for the terms and structures involving taxonomic rank and hierarchy, we referred to the Vertebrate Taxonomy Ontology (VTO) (*Midford et al., 2013*) and imported several of its major classes (with subclasses and all the annotations) in order to demonstrate the functionality of the FO. We also considered the biodiversity standard outlined in the Darwin Core (*Wieczorek et al., 2012*), and other related ontologies such as the Zebrafish Information Network (ZFIN) (*Sprague, 2003*), as the references for the FO vocabulary creation. These ontologies are related to fish, popular in their domain, frequently used, and regularly updated by the research community. Hence, they are the most relevant choices as the main vocabulary provider for fish rank and terms for the FO. As an example, we imported the class "Chordata" and all of the subclasses for the genus *Rastrelliger* and *Chiloscyllium* from the VTO, and reused the terms "Location" and "Taxon" from the Darwin Core in the FO. Some generic terms like "Species" were adopted due to their usage in many popular ontologies. The summary of imported classes is shown in Table 1.

The FO is created with the aim of integration and standardization; thus it is imperative to ensure that the terms in the ontology have a unique identifier (ID) that has not been used in other ontologies. A unique ID for a term allows cross-referencing between related databases and ontologies, without the confusion of same existing terms with different functions. There are many ways to create a unique ID; however, following an example of a globally accepted guideline will ease future integration with the FO. As such, we adopted the guidelines issued by the Open Biological and Biomedical Ontologies (OBO Foundry) (*OBO Technical Working Group, 2016*; *Smith et al., 2007*) and created each term in the FO using an ID which starts with the prefix 'FO' followed by unique digit numbers (e.g., "FO_XXXXXX" where X is a digit).

There are many tools created for ontology validation such as the inference and rule engine. However, it is apparent that human validation is still mandatory in the current state of practice for ontology learning (*Zhou, 2007*). Furthermore, most ontology learning results have mainly been evaluated by domain experts manually. As such, a logical evaluation was conducted by fish experts to verify the naming of concepts and to validate the hierarchy of the terms, which the FO presented. Criteria such as accuracy, complexity, semantic consistency, terms redundancy, naturalness, precision, completeness, and verifiability were
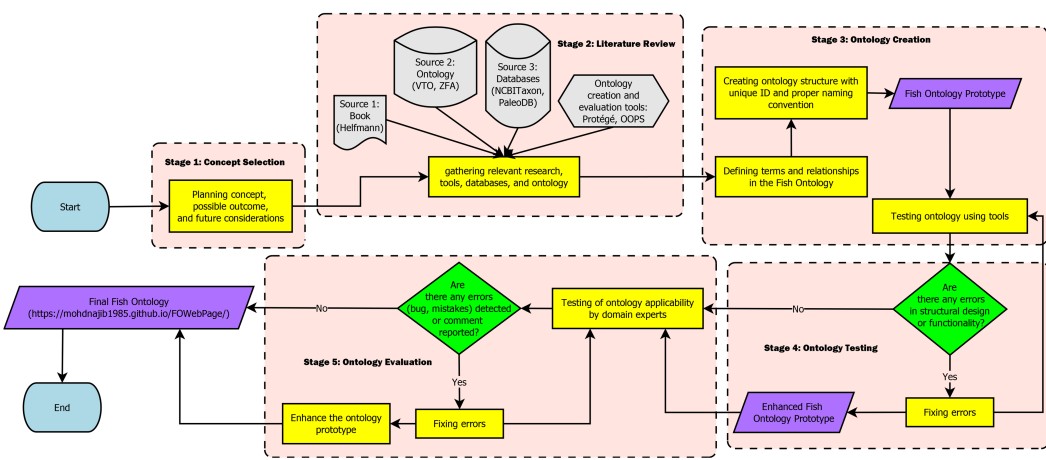

**Figure 1** Fish Ontology workflow.

checked using questions such as "what if we do not know the name of the specimen", "what if we only know its common name", "what if the specimen is similar to certain kind of known specimen", or "what if we were to find a completely unknown specimen".

The workflow for creating the FO in this paper is separated into five stages which are: (1) Concept Selection, (2) Literature Review, (3) Ontology Creation, (4) Ontology Testing, and (5) Ontology Evaluation. Figure 1 shows the workflow of the FO. In the concept selection stage, we first decided on the ontology concept, its possible structure, and future considerations needed to create the ontology. During the literature review, relevant research, such as papers, books, and tools were gathered. Databases and ontologies that are relevant for adoption into the FO were also researched at this stage. Next, in the ontology creation stage, terms and relationship to be used in the ontology were defined and a proper structure of the ontology was created. Additionally, at this stage, the naming convention used for the terms were selected, and any relevant databases and ontologies were imported to the ontology. Subsequently, in the ontology testing stage, the functionality of the FO was tested to detect and fix problems with the terms, structure, or relationships. Finally, during the evaluation, the ontology was evaluated by the fish and biodiversity researchers for its applicability. At this stage, mistakes, bugs, or comments were collected and resolved to improve the ontology.

In this work, we show the applicability of the FO on several areas such as determining if a specimen is a fish, determining the type of fish based on characteristic(s), morphology, name, or taxonomic rank, determining its conservation status (extant or extinct), and determining whether or not it is an ancient species. Examples of its applicability are presented in the 'Results' section.

# RESULTS

## Fish Ontology framework and content

The Fish Ontology proposed in this paper contains 1,830 classes, 27 object properties, 500 species names, with 1,223 synonyms, eight fish groups, and nine fish characteristics. It

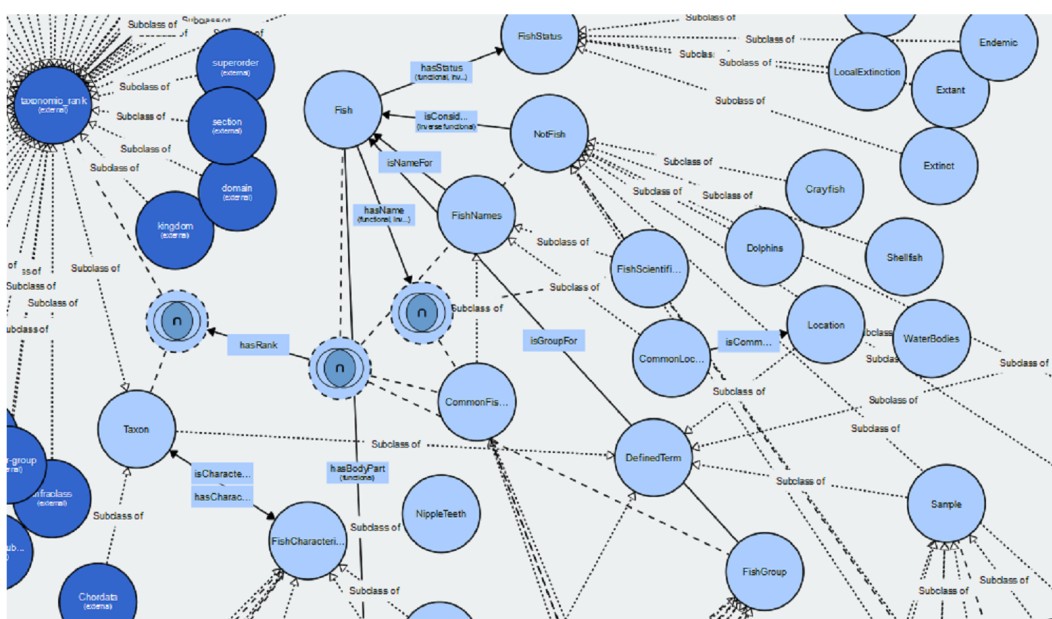

**Figure 2** **The Fish Ontology (FO) architecture.** The architecture shows how the classes are related to each other and to other ontology classes. The dark blue circles represent terms from other ontologies while light blue circles represent terms from the FO.

is the first of its kind to provide automated fish classification based on taxonomic rank, group, name and characteristics. Given that the FO is intended for broad classification of fish, including common extinct ones, the FO is able to classify jawless fish, early jawed fish and living fossil fish. A graphical illustration of several main classes in the FO and its integration with other ontologies such as the VTO is demonstrated in Fig. 2. The online version of the ontology can be accessed at https://mohdnajib1985.github.io/FOWebPage/. The OWL file for the FO with all of its imported classes is available as a supplementary material (Refer to Additional File FishOntology.owl).

The classes in the FO are created as a base for integration with other ontologies mentioned in Table 1, and with any related ontologies that might be useful for fish recognition such as the popular Gene Ontology (GO). For the "Taxon" class, it is organized in single inheritance, up to the species level whenever possible to increase the reasoning capabilities and expand its scope by including relationship(s) and annotation to the terms. This includes imported classes, which are linked to their respective class types. Each FO branch is organized hierarchically by the means of "is_a" (or subclass of) relationship, by appropriately placing it under a single root term. Each class has annotations to enrich its meaning and purpose. Examples of the relationships are shown in Table 2.

The FO contains 253 classes dedicated to fish studies and 38 classes related to fish sampling processes. These classes are well suited for describing sample and specimen related terms. In combination with suitable classes, relations, and annotations, the utility of the FO for automated fish species recognition through sample and specimen data is likely to be improved. Some of the classes such as "FishSampling" and "FishName" are structured in a multiple inheritance structure, with some classes being subclasses of the

**Table 2  Example of relationships in the Fish Ontology.**

| Properties | Explanation | Example |
|---|---|---|
| is_a | A subclass in OWL | Overharvesting is_a CausesOfThreat |
| hasRank (FO:0000097) | Describe a term which has a taxonomic rank | Carpet Shark hasRank of Orectolobiformes |
| isNameFor (FO:0000235) | Describe a name for some other class | FishNames isNameFor Fish |
| isGroupFor (FO:0000171) | Describe a group of some class | FishGroup isGroupFor Fish |
| isPartOf (FO:0000280) | Describe a situation where the class is part of something | PreflexionLarva isPartOf Larva |

**Table 3  Statistics for the Fish Ontology cross references.**

| Resources | Number of cross references |
|---|---|
| NCBITaxon | 264 |
| Teleost Taxonomy Ontology (TTO) | 317 |
| PaleoDB | 1,091 |
| Marine Top Layer Ontology (MarineTLO) | 14 |
| Gene Ontology (GO) | 2 |
| Total | 1,688 |

same class; an example is the class "Trap" which is the subclass of "FishingGear" and "FishSamplingMethod". As aforementioned, most of the new terms were created based on the reference book (*Helfman et al., 2009*) because to the best of our knowledge, there are no suitable ontologies from which we could import these classes, while some of the terms that we found were poorly defined and structured. However, we have included cross-references of several classes for potential mapping to relevant external resources, including the FishBase, Teleost Taxonomy Ontology (TTO), and National Centre of Biotechnology Information Taxonomy Database (NCBITaxon) (*Froese & Pauly, 2000*; *Midford et al., 2010*; *Federhen, 2011*). Table 3 shows the statistics of cross referencing of the FO classes to other resources.

## Inference capabilities

We have created relationships which allow a specimen (and sample) to be inferred and automatically analyzed in the areas of fish grouping, taxonomic rank, and common fish names. We focused most of our modelling activities on these aspects. The specimen (and sample) which is not inferred would only be shown as subclasses of "Sample" or "Specimen" classes; however after being inferred using the reasoner provided by Protégé, each one of them will be properly classified according to their respective parameters. Furthermore the inferred results can show which individual shares the same trait(s) as the sample and suggest what kind of group it fits into based on its characteristics.

The FO also provides a structure to determine whether a specimen or a species is actually a fish or otherwise by using the reasoning capability. Figure 3 shows the results of the inferring process which demonstrates the classification of specimen as a fish or otherwise, and what group it belongs to in the taxonomic hierarchy. Infer Result A shows how a specimen (Specimen5) is recognized by the reasoner as a "Whale" and leads the
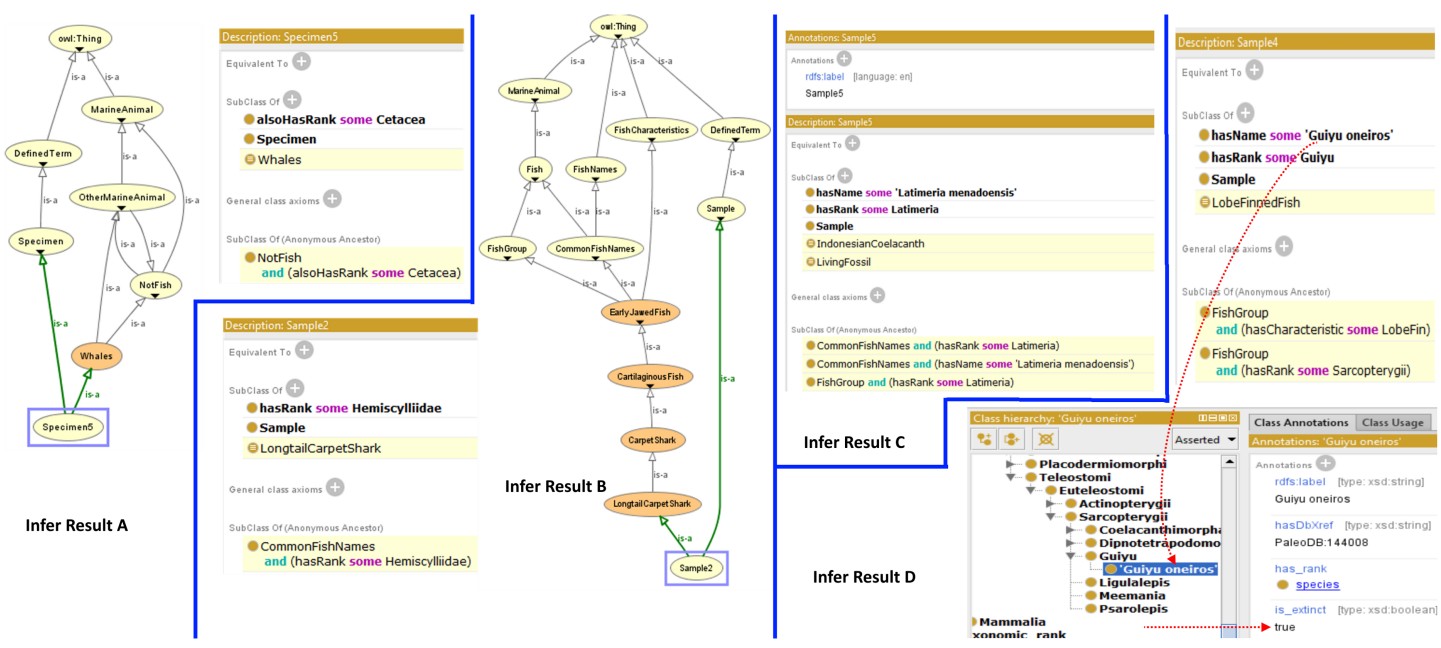

**Figure 3** Results from the Fish Ontology inferring process.

reasoner to recognize it as "OtherMarineAnimal" and "NotFish". Infer Result B shows how a sample (Sample2) is recognized as a "LongtailCarpetShark", which leads the reasoner to recognize that it is a fish. Infer Result C shows that Sample5 is recognized as a "LivingFossil" while Infer Result D shows that Sample4 is actually a species called *Guiyu oneiros* and subsequently recognized as an extinct species.

Sample queries are presented in Fig. 4 using SPARQL (Query A) and SPARQL-DL (Query B). Improved results were obtained using the SPARQL-DL query, which could query inferred data in the ontology compared to a SPARQL query. As shown in Fig. 4, new classes were found in Query B results, which are obtained from the imported class and integrated terms from other ontologies. The results in Fig. 5 show how additional data can be retrieved from the FO using the DL query.

## Evaluation

To evaluate the quality of the Fish Ontology, we follow the Gruber method for ontology construction (*Gruber, 1995*). There are five criteria highlighted which are clarity, coherence, extendibility, minimal encoding bias, and minimal ontological commitment. To measure the clarity level of the FO, the ontology definitions should be objective and independent of the social and computational context. In FO, all the definitions are stated in such a way that the number of possible interpretations of a concept would be restricted. The clarity test results for the FO are divided into five parts, which are:

1. No Cardinality Restriction on Transitive Properties
2. No Meta-Class
3. No Subclasses of RDF Classes

```
Snap SPARQL Query:

PREFIX rdf: <http://www.w3.org/1999/02/22-rdf-syntax-ns#>
PREFIX owl: <http://www.w3.org/2002/07/owl#>
PREFIX rdfs: <http://www.w3.org/2000/01/rdf-schema#>
PREFIX xsd: <http://www.w3.org/2001/XMLSchema#>
PREFIX fo: <http://mybiodiversityontologies.um.edu.my/FO.owl#>

SELECT * WHERE {
fo:Sample1  rdfs:subClassOf ?sub.
}
```

|  | ?sub |
| fo:Sample1 |  |

**Query A**

```
Snap SPARQL Query:

PREFIX rdf: <http://www.w3.org/1999/02/22-rdf-syntax-ns#>
PREFIX owl: <http://www.w3.org/2002/07/owl#>
PREFIX rdfs: <http://www.w3.org/2000/01/rdf-schema#>
PREFIX xsd: <http://www.w3.org/2001/XMLSchema#>
PREFIX fo: <http://mybiodiversityontologies.um.edu.my/FO.owl#>

SELECT * WHERE {
fo:Sample1  rdfs:subClassOf ?sub.
}
```

|  | ?sub |
| fo:CartilaginousFish |  |
| fo:Sample |  |
| owl:Thing |  |
| fo:FishNames |  |
| fo:DefinedTerm |  |
| fo:EarlyJawedFish |  |
| fo:Fish |  |
| fo:FishGroup |  |
| fo:CommonFishNames |  |
| fo:Sample1 |  |

**Query B**

**Figure 4   A sample of query to check the inferred results.** Results from Query A (using SPARQL) were retrieved before the inferring process, while results from Query B (SPARQL-DL) were retrieved after the inferring process.

4.  No Super or Sub-Properties of Annotation Properties
5.  Transitive Properties cannot be Functional.

Results for clarity test 1 and clarity test 5 are shown in Fig. 6 below. Since fish data are large in volume, there is a need to add more data to the FO over time. As such, there is no cardinality restriction assigned to any transitive properties in the FO. Figure 6 also shows that the transitive properties in the FO are not functional because it relates to more than one instance via the property. As for clarity test 2, clarity test 3 and clarity test 4, Figs. 7

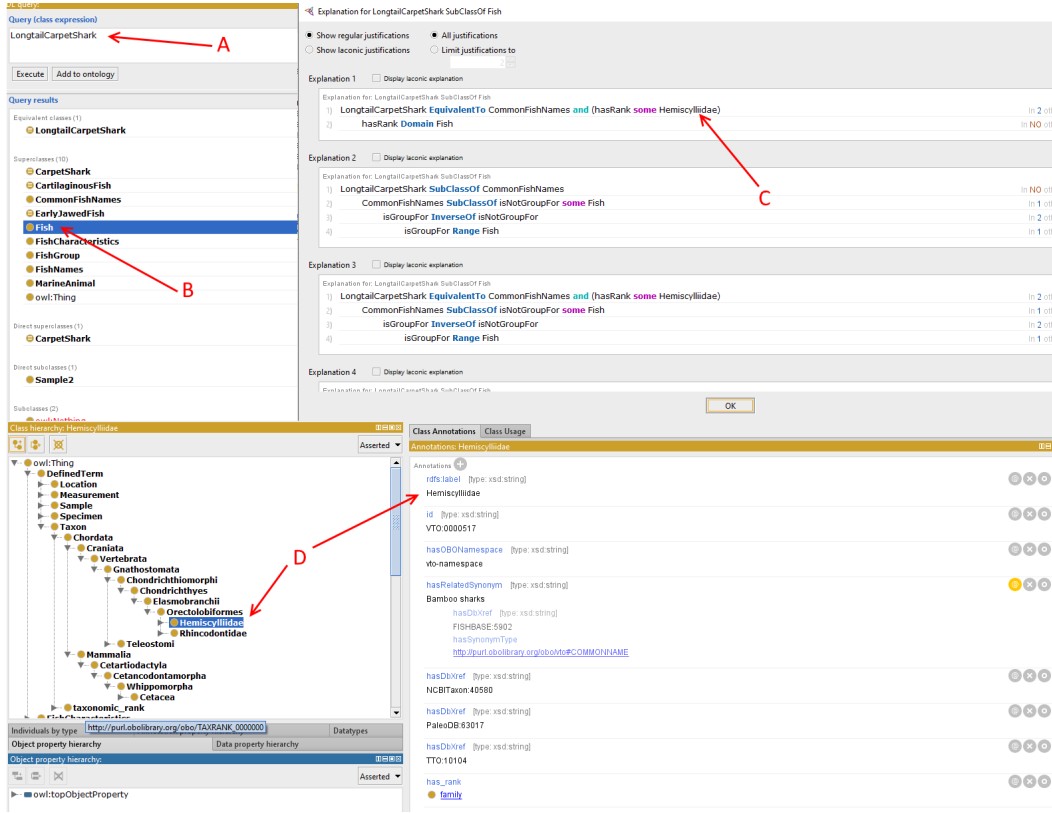

**Figure 5  Types of information obtained from the Description Logic (DL) query.** The DL query shows how a long tail carpet shark is inferred in the DL query (A). In (B), the shark is inferred as Fish. In (C), the DL query shows what kind of fish it is while in (D), the shark rank in the fish taxonomic structure is subsequently inferred.

and 8 show that there are no meta-classes, no properties with a class as a range, and no sub-classes of RDF classes in the FO. All the five clarity tests are automatically performed in the latest Protégé version.

To ensure the coherence quality of the FO, the definitions of concepts given in the ontology as well as the inferences drawn from the ontology must be consistent with its definitions and axioms. Based on our evaluation, most of the inferred terms from the FO appeared to be consistent with its definition and axioms. As an example, in Fig. 3 when the FO inferred that specimen5 is a whale, it also inferred that it is not a fish, and it also showed the correct taxon rank. The formal part of the FO is checked by following the five consistency criteria listed below and ensuring that all return true:

1. Domain of a Property should not be empty
2. Domain of a Property should not contain redundant Classes
3. Range of a Property should not contain redundant Classes
4. Inverse of Symmetric Property must be Symmetric Property
5. Inverse Property must have matching Range and Domain.

Protégé forces the user to always be wary about an empty domain, redundant classes, and properties. As such, coherence tests 1 to 3 are achieved and can be further viewed
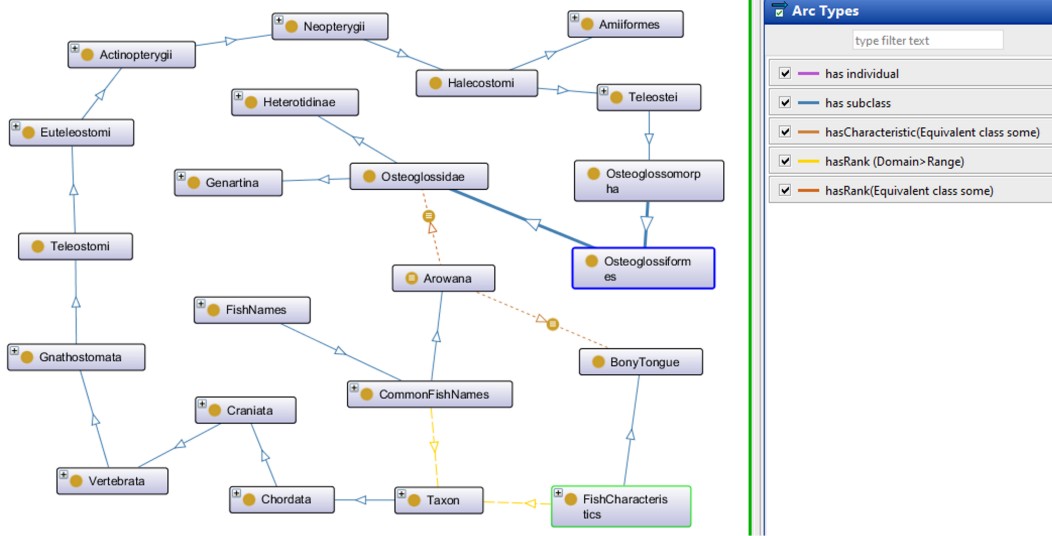

**Figure 6  Results for clarity test 1 and clarity test 5.**

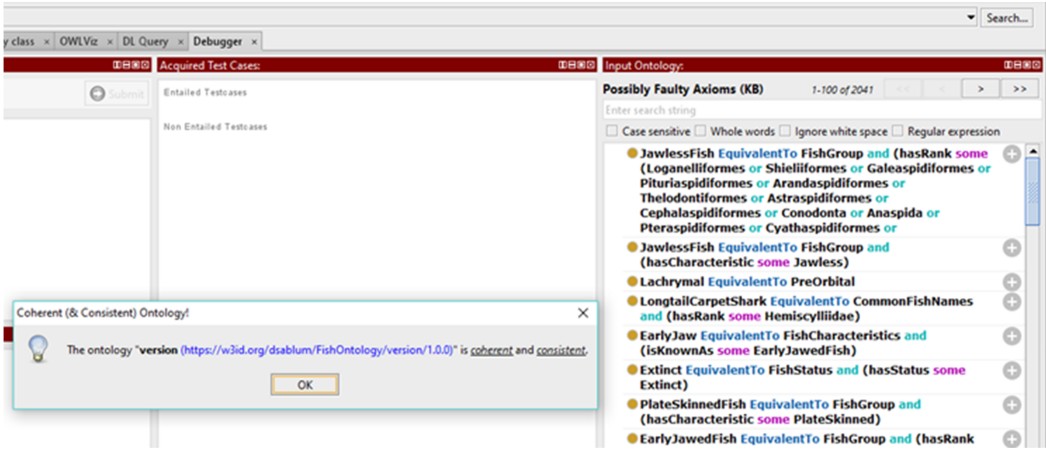

**Figure 7  Results of the coherence test using Protégé Ontology Debugger tool.**

via the ontology itself. For coherence test 4, we provide an example of the property "isSimilarTo". The class "CosmoidScales" is related to the class "PlacoidScales" via the "isSimilarTo" property. Subsequently we can infer that "PlacoidScales" is also related to "CosmoidScales" via the "isSimilarTo" property. Figure 7 shows the results of coherence test using the Ontology Debugger Tool from Protégé. The coherence test from this tool checks for possible faulty axioms. The ontology passed the coherence test provided by this tool. Figure 8 shows the results for test 5 displaying that the properties "hasCharacteristic" and "isCharacteristicFor" have matching range and domain.

An ontology should be extensible which means allowing addition of new concepts, according to the current development in the field (*Barbosa & Da Silva, 2001*). In this paper, the FO is made extensible via the design consisting of concepts, classification
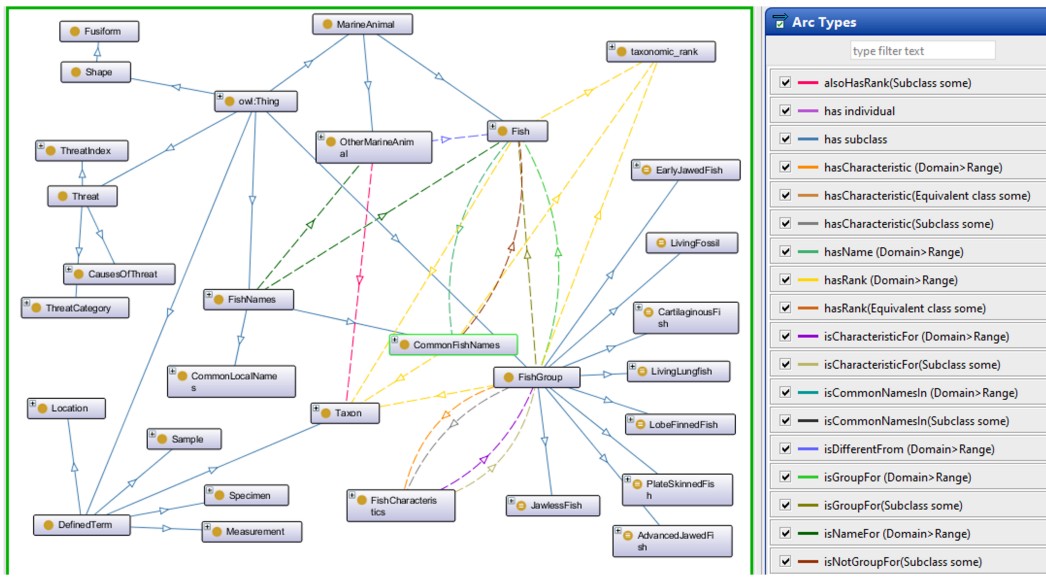

**Figure 8** Results for clarity tests (2, 3, and 4) and coherence test (5).

hierarchy represented as classes, from general to specific. Applying reasoning to the FO helps to define new concepts (generated from an ontology) from defined generic concepts (books and other databases). As such, the FO place any related concepts derived from other generic concepts in its class hierarchy to represent information that defines a specimen. Classes and annotations that may be useful for future integration such as genetic are added, since such content will further enhance FO's extendibility. Tables 1 and 3 show all the classes that are linked or cross-referenced by the FO to demonstrate the extendibility of the FO. Since the first FO design, we have considered integrating terms from other ontologies into the FO. By placing any related concepts derived from other generic concepts in its class hierarchy, the FO represents information that defines a fish specimen, linking it with terms from other ontologies.

It is a preferred practice to make an ontology which would require minimal ontological commitment so that it is sufficient to support the intended knowledge sharing (*Man, 2013*). Ontology modelers sometimes have a vague idea of the role each concept will play such as their semantic interconnections, within the ontology. If necessary, they can annotate new development ideas during subsequent ontology updates (*Nicola, Missikoff & Navigli, 2005*). As such, an ontology should make as few claims as possible about the domain while still supporting the intended knowledge sharing. By reusing existing concepts from books, databases and other ontologies on fish, the FO has low ontological commitment, making it more extendible and reusable (*Freitas, Stuckenschmidt & Noy, 2005*). Also, since most of the new concepts introduced in the FO are from books and published journal articles, they are likely to be more understandable and preferred among the user community (*Helfman et al., 2009*; *Last et al., 2010*; *Chong, Lee & Lau, 2010*).

Encoding bias occurs when a representation choice is made for the convenience of notation or implementation. By minimizing encoding bias, knowledge-sharing agents

may be implemented in different representation systems and styles of representation. An ontology that is independent of the issues of implementing language is considered to have minimal encoding bias. Also, the conceptualization of the ontology should be specified at the knowledge level and must be independent of symbol-level encoding. As shown in Tables 1–3, encoding bias in the FO is reduced by the choices of using OWL as the representation language, and of adopting terms from books, database, and related ontology. As demonstrated in Fig. 7, the Protégé Ontology Debugger tools have fully examined all possible encoding biases in the ontology and have cleared the FO as coherent and consistent.

To strengthen the results of the FO evaluation, we use an online ontology evaluation tool named OntOlogy Pitfall Scanner! (OOPS!) (*Poveda-Villalón, Gómez-Pérez & Suárez-Figueroa, 2014*). OOPS! uses a checklist to ensure that best practices of ontology creation are followed and that the bad practices are avoided. The inventor created a catalog of bad practices and automated the detection of as many of them as possible (41 currently). The evaluation of the FO using the OOPS! tools is shown in Fig. 9. There are 1,794 cases listed in the minor pitfall categories, 19 cases in four important pitfall categories, and 11 cases in four critical pitfall categories. Compared to the ontology debugger tools in the Protégé, there are many error flags that can be found in the FO by using OOPS!. However, most of them are minor, and the important and critical pitfalls problems are mostly caused by the same features in the FO, and is further elaborated in paragraphs 6 and 7 in the 'Discussion' section.

## DISCUSSION

In this paper, we developed a Fish Ontology framework which is a general-purpose ontology that allows integration of fish related ontologies containing standard terms and relationships. The design of the FO is flexible enough to accommodate any ontology containing data or knowledge about fish. Even in cases where integration can be difficult, the FO can be tweaked in order to incorporate new biodiversity related ontology. One example is linking the FO to the MarineTLO which is an upper level ontology for marine species (*Tzitzikas et al., 2016*). The MarineTLO does not have a class named "Fish" that can map to data from the FO; however since the MarineTLO provides classes of taxonomic rank such as "Species" and "Genus", and related classes such as "MarineAnimal" and "Specimen", the FO provides classes and annotations to link to these classes. The same steps can be done with the ZFIN, which contains "zebrafish anatomical entity" and "Stages" as main classes. The FO provides complementing classes to match the classes provided by the ZFIN such as "FishAnatomicalEntity" and "OtherStagesTerminology".

The FO is able to prepare captured and observed fish specimen data, mapped and structured in a way that the meaning is expressed in a machine understandable format. Since data representation in the form of an ontology allows the information to be linked by using Semantic Web applications, we envision several practical cases of real life applications using this ontology. As shown in the results, the FO can infer conservation and evolutionary statuses of a fish as well as show related characteristics, e.g., early jawed fish, which are

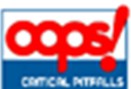

Figure 9 Results of the Fish Ontology evaluation using the OntOlogy Pitfall Scanner tool (*Poveda-Villalón, Gómez-Pérez & Suárez-Figueroa, 2014*).

useful information for interested museum visitors. The FO's ability to infer location and habitat of the fish can be useful for students or researchers. They can use the FO to identify species using local names, since all fish names in the FO are linked to other database repositories. Linkage of the FO to other ontologies via reusing of terms allows the search for relevant information such as genetic data of a specific fish species. In this way, the FO is able to produce new knowledge which is useful to biologists.

The current version of the FO can utilize specimen grouping and characteristics to determine whether a specimen is a fish or otherwise, provide taxonomic information and heredity of a characteristic rank, and determine conservation status, evolutionary status (ancient or modern) and type (jawless fish is an ancient species). The power of the FO lies in its ability to automate group classification, and ability to link the terms used by fish domain researchers, and other researchers outside the domain. This version uses simple character classification where the user provides the necessary character for the specimen. As an example, the user can specify that "Sample 1 has the characteristic of Plate Skinned", and manually add the characteristic of "Plate Skinned" into the FO. We believe the ideal FO version should also contain anatomical and phenotype data from several classes in the ontology such as "AnatomicalCharacters", "MeristicCharacters", "MolecularCharacters",

and "Morphometric Characters" and these features will be included in the near future. These classes can be useful for pattern recognition, and species taxon recognition studies.

In general, we extract information such as synonym, name, fish grouping, group rank, fisheries, and other fish-related terms from Helfman et al. to form the general structure of the FO. We adopt the usage of the VTO for taxonomic hierarchy, taxonomic related information, and terms related to taxonomic rank. In most cases, the taxonomic structure of the VTO is followed as it is a regularly updated ontology. However, there are exceptions to this, such as the class "Mammalia" which is placed as a subclass of "Sarcopterygii" in the VTO. Although this classification is consistent with the cladistics standpoint that mammals are derived from fish, explicit classification of mammals under Sarcopterygii in the FO would result in the erroneous recognition of a fish-like mammal, e.g., whale, as both a fish and a mammal. We have therefore placed Mammalia under the higher taxonomic rank of Chordata and made annotation within the FO to highlight this choice.

The NCBITaxon, an automatic translation of the NCBI taxonomy database into OBO or OWL format (*Federhen, 2011*), is also used by the FO as a secondary source for terms regarding taxonomic rank. Both the VTO and NCBITaxon differ in hierarchical organization and definitions. One of the most distinctive feature of the VTO compared to the NCBITaxon is its broad taxonomic coverage of vertebrates. The NCBITaxon excludes many extant and nearly all extinct taxa while largely includes only species associated with archived genetic data, complemented by data from the PaleoDB and the TTO to provide an authoritative hierarchy and a richer set of names for specific taxonomic groups (*Midford et al., 2013*). Therefore the VTO is more relevant to the FO's purpose for a comprehensive fish taxonomy information, since the VTO is built based on several taxonomic resources including the NCBITaxon the Paleobiology Database (PaleoDB), and the Teleost Taxonomy Ontology (TTO) (*Alroy, Marshall & Miller, 2012*; *Dahdul et al., 2010*). Having said that, any taxonomic ranks covered by the NCBITaxon but are not covered by the VTO, such as the species *Protanguilla palau* and the subfamily Oxudercinae are incorporated in the FO to improve coverage of fish data. More examples on the differences between the main reference book, the VTO, and the NCBITaxon, as well as what the FO uses are shown in Table 4.

New knowledge emerges every day so there is a need to add new concepts and relationships to the existing ontology. Proposing new vocabulary in biodiversity is not uncommon, since ontologies in this domain are presently insufficient and many are under development. Available standard vocabulary is not comprehensive enough to cover all the terms needed to make an ontology in the fish domain. In most cases, new terms must be proposed based on the rationale utilized in the ontology. One such example is that of Hymenoptera Anatomy Ontology, where new terms had to be proposed to expand the ontology (*Seltmann et al., 2012*; *Seltmann et al., 2013*). It should be possible to extend an ontology without altering the existing definitions. As such, the need for easy ontology extension is prioritized while creating the FO. The new terms are checked for its suitability to be adopted as a standard vocabulary for fish scientists. The use of adopted terms and concepts from our main references is further clarified with domain experts (Amy Y. Then,

**Table 4  Term adoption example in the Fish Ontology.**

| Term example | *Helfman et al. (2009)* | VTO (*Midford et al., 2013*) | NCBITaxon (*Federhen, 2011*) | Fish Ontology (FO) |
|---|---|---|---|---|
| Furcacaudiformes (order) | Classified as Subclass of Thelodonti (superclass). | Classified as subclass of Agnatha (class). | Not classified. | Follows and reuses the VTO terms. |
| JawlessFish | Contains species and information for jawless fish species. | No classes and annotations found, but related species are classified. | No classes and annotations found, but related species are classified. | Follows *Helfman et al. (2009)* for labeling. |
| LobeFinnedFish | Classify as Sarcopterygii (page 4). | No classes and annotations found, but related species are classified. | Classified as Coelacanthiformes. | Follows *Helfman et al. (2009)* for classification and labeling. |
| Gobiidae (family) | Listed and classified as family. | Listed and classified as family. | Listed and classified as family. | Follows and reuses the VTO terms. |
| Oxudercinae (subfamily). | Not listed. | Not listed. | Classified as a subclass of Gobiidae (family). | Follows and reuses the VTO classification up to the lowest existing taxonomic terms covered (Family Gobiidae). Adopts NCBITaxon terms for Subfamily Oxudercinae onwards. |

Chong V. Ching) in order to represent and map the appropriate contents to reflect the diverse aspects of fish (*Helfman et al., 2009*).

Regarding ontology evaluation, there are reasons a number of errors were flagged by the OntOlogy Pitfall Scanner (OOPS!) but none can be detected by using the tools from Protégé. The most apparent reason is because the scope of evaluation for both methods are different. In Protégé, only the classes and its relationship structures created in the ontology are being evaluated, while in OOPS!, the classes, relationships, mapping and future integration problems are being evaluated, giving different results. A number of errors detected by OOPS! can be attributed to the important FO feature of reusing terms from other ontologies in order to reduce redundancy in global usage. As mentioned earlier, terms and structures taken from other ontologies have their own unique ID and metadata to indicate associated function. However, since most of these terms are directly used in the FO, the OOPS! tool flagged these occurrences as critical errors such as "P24: using recursive definitions", "P32: several classes with same labels", and "P40: namespace hijacking".

Other pitfalls such as P02, P04, P08, P11, P13, P30, P36, and P41 (refer to Fig. 7) are considered acceptable since there are constantly new items to be added to the ontology along with the necessary annotations, relations and property constraints. As an example, errors flagged under pitfall "P19: defining multiple domains or ranges in properties", is due to the modeling of FO for increased inference capacity (*Poveda-Villalón, Gómez-Pérez & Suárez-Figueroa, 2014*). In a typical ontology, inferring capabilities is used to discover new relationships. In our work, we used inferring capabilities for automated fish species recognition. As such, we had to avoid using 1 to 1 relationships for the domain and range, instead we expanded the domain and range of each property for a more reliable automated

species discovery. Although OOPS! flagged these property expansions as pitfall errors, we deem these as minor for the purpose of the FO development.

The FO covers the terms for fish domain which are not well described by other ontologies, such as terms used for fish studies ("maturity", "age") or sampling experiment ("weight", "length"), particularly those related to automatic classifications, annotations, and relationships. There are however some terms in the FO created using parameters rarely used outside of this domain, such as "FishDatabases" which are for any known databases for fish, or "GasBladder" which is a specific organ for "Actinopterygii". The differences between the FO and other fish related ontologies and databases is its ability to provide automated classification of unknown specimen.

There has been efforts to create ontologies for recognition purposes such as Hymenoptera (*Balhoff et al., 2013*); however, in the fish domain ontologies were created to focus more on classification rather than recognition, such as the MarineTLO, NFO, TTO and the ZFA. In this paper, the FO was created to focus on automated fish recognition. The comparison of FO with other related ontologies in the fish domain is presented in Table 5. For the purpose of this paper, we considered ontologies which are most related to Fish Ontology while not taking into consideration systems that use these ontologies as their underlying framework. FishBase was included as it is the most referred portal in this domain (600,000 Visits/Month).

Development of the FO for classification of several highly diverse groups, such as bony fishes, advanced jawed fish, sharks, skates, and rays, is an ongoing effort.

We envision the FO to expand by incorporating additional components such as fish models, fisheries parameters, gene annotations and other relevant information as aforementioned. These parameters will further enhance fish recognition capabilities to recognize fish based on physical features or gene annotations. We will focus on parameters that influence the grouping process such as shape and characteristic recognition, and anatomical metric distinctions. Other than including more terms and defined relationships, we are considering to increase granularity by linking to other relevant and established ontologies, such as the Gene Ontology (GO), the Zebrafish Information Network (ZFIN), the Vertebrate Skeletal Ontology (VSAO), and the Protein Ontology (PO) (*Ashburner et al., 2000*; *Sprague, 2003*; *Dahdul et al., 2010*; *Natale et al., 2011*). In the near future we aim to integrate the FO with other ongoing efforts in our research group such as the Otolith Ontology, Monogenean Ontology, and the Monogenean Haptoral Bar Ontology (MHBI) (*Abu et al., 2013a*; *Abu et al., 2013b*). There is also consideration to link related ontologies to existing fish databases using the FO as a mediator (*Great Lakes Fishery Commission, 1997*; *iBOL Working Group, 2005*; *International Game Fish Association, 2016*; *National Institute of Water and Atmospheric Research, 2016*; *Froese & Pauly, 2016*; *Shao, 2001*). Furthermore we also hope to evolve the FO so that in the future, our other ongoing works on different type of fish related recognition tools or technique can be applied to enhance its inferencing capabilities (*Abu et al., 2013a*; *Leow et al., 2015*; *See et al., 2016*; *Salimi et al., 2016*; *Wong et al., 2016*; *Kalafi et al., 2016*).

The annotation of fish and fisheries resources in the FO and other related ontologies is a response to the emerging need for data sharing and integration especially for fish

**Table 5** Differences between Fish Ontology with other related ontology and database.

| | FishBase | MarineTLO | NFO | FO |
|---|---|---|---|---|
| Domain coverage | Fish and fisheries | Marine life | Fisheries | Fish |
| Ontology based | No | Yes | Yes | Yes |
| Underlying sources | 33,500 Species, 319,000 Common names, 58,100 Pictures, 53,800 References information from the FishBase Consortium and 2,270 Collaborators | FLOD (Fisheries Linked Open Data), ECOSCOPE (A Knowledge Base About Marine Ecosystems), WORMS (World Register of Marine Species), DBpedia, and FishBase | ISSCAAP (International Standard Statistical Classification of Aquatic Animals and Plants), AGROVOC (a portmanteau of agriculture and vocabulary) thesaurus, ASFA (Aquatic Sciences and Fisheries Abstracts) thesaurus, and FIGIS (Fisheries Global Information System) data | TTO, NCBITaxon, and VTO (with linked information from FishBase and PaleoDB) |
| Fish information provided | Common Name, Scientific Name (both species and genus, and species id), Information by Family, by country/island, by ecosystem, or by specific topic | Species, Scientific Names, Common Names, Predators, Authorships, Ecosystems, Countries, Water Areas, Vessels, Gears, EEZ, Bibliography, Statistical Indicators | Imported data sources in the owl file cover the topic of water areas, species taxonomic classification, ISSCAAP commercial classification, Aquatic resources, Land areas, Fisheries commodities, Vessel types and size, Gear types, AGROVOC data and ASFA data. | Species, Taxon Information, Fish Name, classes related to fish studies and fisheries |
| Difference in fish searching concept | When searching for a fish species in FishBase, details such as names (common, scientific, other language), taxon classifications, environment, climate, range, distribution, size, weight, age, short description, biology, life cycle, mating behavior, main references, IUCN red list status, threat to human, and human uses will be provided (if available). Furthermore, other information such as the species countries, FAO areas, occurrences, ecology, genetics, internet sources, special reports, tools, and xml data sources are available as additional information sources. | Searching a fish species through the MarineTLO owl file is not possible. However its competency query v4 suggested that it covers a wide range of search topics such as species and its scientific name, its WORMS classification, prey and predator information, references, images, general terms, identifiers, competitors, biotic type of predator, assignment data, its biological environment, common name with complementary information, and water areas with their FAO codes. | Searching a fish species through the NFO owl file is also not possible. However it's imported data sources suggested the you can get information on fish species' ISSCAAP classification, ASFIS list (covers names and extensive details of species taxonomic rank), Aquatic Sciences and Fisheries Abstracts (ASFA) bibliographic database ( links to FAO Fish Finder Fact Sheets which cover synonyms, FAO names, scientific names with original description, diagnostic features, Geographical distribution, habitat and biology, size, interest to fisheries, local names, source of information and Bibliography) | When FO search for a fish, it provide its taxon information, scientific name, common name, synonym, and links to TTO, FishBase and PaleoDB (if available). When unknown species is inferred in the FO, it can find whether a specimen or a sample is a fish or not fish, providing its taxon rank, full name, its characteristic, grouping, and its extinction status. Future concepts will allows it to provide data on fish morphology, genetic content and other fish species related information such as country maturity and other related information (like FishBase). FO infers the type of fish based on parameters provided |

data resources (*Ashburner et al., 2000*; *Gangemi et al., 2004*; *Bizer et al., 2009*; *Dahdul et al., 2010*; *Dahdul et al., 2012*; *Midford et al., 2010*; *Midford et al., 2013*; *Federhen, 2011*; *Natale et al., 2011*; *Schriml et al., 2012*; *Tzitzikas et al., 2013*; *Van Slyke et al., 2014*; *Pesquita et al., 2014*) and will be highly relevant for the future of fish and fisheries related research.

# CONCLUSION

An ontology for the fish and fisheries domain with automated fish recognition is introduced and discussed in this paper. The Fish Ontology (FO) is a new ontology with the feature of taxonomic-based recognition of fish by importing existing ontologies related to fish such as the VTO, ZFA, and TTO. The ontology infers information based on criteria such as names, rank, or characteristics, thus allowing recognition from specimen characteristics. The base terms are taken or imported from related ontologies or naming standards which enhanced the FO's fish recognition and cross-referencing capabilities. The potential usage of the ontology is huge, especially as a comprehensive information provider for interested users such as fishermen, museums, restaurants, or for research purposes. More importantly, the FO could be used as a framework to build Semantic Web systems for data integration to be applied in biodiversity research in the fish and fishery domain.

## Funding

This research was supported by the Ministry of Higher Education Malaysia's Fundamental Research Grant Scheme (FP032-2014B), University Malaya's Grant (BK018-2015) and University of Malaya's Postgraduate Research Grants (PG130-2013A, PG212-2014B). The funders had no role in study design, data collection and analysis, decision to publish, or preparation of the manuscript.

## Grant Disclosures

The following grant information was disclosed by the authors:
Ministry of Higher Education Malaysia's Fundamental Research Grant Scheme: FP032-2014B.
University Malaya's Grant: BK018-2015.
University of Malaya's Postgraduate Research Grants: PG130-2013A, PG212-2014B.

## Competing Interests

The authors declare there are no competing interests.

## Author Contributions

- Najib M. Ali conceived and designed the experiments, performed the experiments, analyzed the data, wrote the paper, prepared figures and/or tables, reviewed drafts of the paper.
- Haris A. Khan and Manas Gaur wrote the paper, reviewed drafts of the paper.
- Amy Y-Hui Then and Chong Ving Ching analyzed the data, wrote the paper, reviewed drafts of the paper.

- Sarinder Kaur Dhillon conceived and designed the experiments, performed the experiments, analyzed the data, contributed reagents/materials/analysis tools, wrote the paper, prepared figures and/or tables, reviewed drafts of the paper.

## Data Availability

mohdnajib1985. (2017 August 29). mohdnajib1985/FOWebPage: Fish Ontology. Zenodo. 10.5281/zenodo.852711.

## Supplemental Information

Supplemental information for this article can be found online at http://dx.doi.org/10.7717/peerj.3811#supplemental-information.

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
