# Peer review of "Fish Ontology framework for taxonomy-based fish recognition"

_PeerJ, doi:10.7717/peerj.3811_

## Round 0.1 · original submission · Minor Revisions

A very good piece of work. Nice comments from two reviewers and suggestions for minor revisions. please do as per the reviewers comments.

Reviewer 2 provided an annotated pdf file, so please carefully check all the suggestions.

·

Basic reporting

The English is fully comprehensible, but relatively basic and there are small errors throughout the text, eg. in the Abstract "The automated classification for unknown specimen is a feature not existing in other known ontologies." It needs some language revision/editing.

Reference list is quite good

Experimental design

No comment; I am not familiar with the more intricate aspects of ontology construction

Validity of the findings

I have been reading this manuscript as one of the managing group of one of the major fish databaes , as editor, of fish papers, and as specialist on fish identification.

Ontologies were very hot when Web 2.0 (the semantic web) was launched but after that there has been very limited progress. Particularly in fish systematics, the hierarchical system already in place is so self-contained that it needs no further elaboration; but at the same time ranks, names and diagnoses shift with advancing research. In systematics, including taxonomy, there are no fixed biological entities, and there is no common nomenclature for body parts. Part of the problem of building a descriptive ontology based on physiology and morphology is the lack of homology between structures with the same names, and for fish the fact that it is not a monophyletic group, so even slight homology requirements for terms is difficult. There are two examples in the MS that can be highlighted. The authors complain that Mammals fall under Sarcopterygians, but in fact mammals are sarcopterygians, and among sarcopteryians usually considered closest to some group of dipnoans. In the table below line 430 there is confusion about "jawless fishes". Actually Agnatha (lampreys, hagfish and ostracoderms) is not a monophyletic groip. Some taxa are actually stem gnathostomes (jawed fishes), which means the term is rather useless. In the same table there are confusions also about lobe-finned fish, which is the same as Sarcopterygii and not part of the Actinopterygii (the sistergroup) and not the same as Coelacanthiformes (which is a subgroup). These and many examples in the term list suggest considerable challenge for a wider fish ontology if it aims at any degree of precision. This is why ontologies tend either to be vey specialized+detailed (e.g. Zebrafish), or at very high level and not very precise (VTO). Trying to combine several ontologies into one, based on a popular book about fish just does not seem right to me. I would probably rather like lo build the ontology for the intended purpose and keep it resticted. Thus I see opportunities in FishBase (potential main adopter of a wider fish-related ontogeny) and GBIF for automatic mashup and crosslinking that could enhance the usefulness of the databases.

Some of the problems with "floating" systematics are adressed in Balhoff, J.P. et al. 2013. A semantic model for species description applied to the ensign wasps (Hymenoptera: Evaniidae) of New Caledonia. Systematic Biology, 62: 639-659. It might be worth considering for adaptation to fishes. At least Hellman et al., already long outdated in the systematic portion is not a viable platform for a precise classification of fish anatomy and morphology or taxonomy.

Consequently, looking at the usefulness, from a practical perspective, I am skeptical that this is ripe for semantic web designs, but it is interesting to look at.

The core novelty is supposed to be the automated detection of unidentified specimens; which should be exactly equivalent to automatic detection of identified specimens. I doubt this is possible because for morphological identification it requires considerable diagnostic detail. I am frequently examining specimens that I am unable to identify because the published information about species in the same genus or family is insufficient or erroneous. There has been several attempts at automatic identification (or classification) of fish species using images, and there exists data from geometrical morphology that could be used for the purpose, but the application is very limited. Fish identification today is made by a combination of morphology and DNA (principally barcoding). For identification purposes there is a range of approaches available (Fischer, J. ed. 2013. Fish identification tools for biodiversity and fisheries assessments: review and guidance for decision-makers. FAO Fisheries and Aquaculture Technical Paper No. 585. Rome, FAO. 107 pp.) The FAO paper highlights the needs and options for fish identification and characterisation under many real-life conditions.

I am not sure present ontology offers competition over conventional databases in providing morphological information on fish species. Both fails probably in adressing sex dimorphism and ontogeny, and natural language descriptions still hold power.

The computational novelty of the FO is combining several related ontologies. But by basing them on a popular treatise on fishes and not providing a solution for shifting views of homology and taxonomy, I am not sure this is the way to go

Reviewer 2 ·

Basic reporting

The paper is written in clear and professional English. There is sufficient literature reference provided. The results are self-contained and relevant to the research question. Figures and tables are professionally structured. Raw data is shared.

Experimental design

The paper's original primary research falls within the aims and scope of the journal. The research is well defined and relevant to the journal gap. The author have clearly identified a knowledge gap and the research fills it.

The research is performed to a high technical standard to answer the research question. Methods used are described in a sufficient details to permit replication.

Validity of the findings

The data in the paper is robust and controlled.

Negative results are accepted and explained.

Additional comments

The authors have presented an important and novel research paper. There are some clarification required, as stated in the comments in the paper.

Annotated reviews are not available for download in order to protect the identity of reviewers who chose to remain anonymous.

---

## Round 0.2 · accepted · Accept

Thank you for addressing all the suggestions and comments by the reviewers.

I hope this work will enrich the taxonomy-based fish identification and the researchers in the field of taxonomy will be benefited.